# The Strong Enhancement of Electron-Impact Ionization Processes in Dense Plasma by Transient Spatial Localization

**DOI:** 10.3390/ijms23116033

**Published:** 2022-05-27

**Authors:** Jiaolong Zeng, Chen Ye, Pengfei Liu, Cheng Gao, Yongjun Li, Jianmin Yuan

**Affiliations:** 1College of Science, Zhejiang University of Technology, Hangzhou 310023, China; 2Department of Physics, College of Liberal Arts and Sciences, National University of Defense Technology, Changsha 410073, China; yechen13@nudt.edu.cn (C.Y.); feipeng1050@163.com (P.L.); gaocheng@nudt.edu.cn (C.G.); liyongjun13@nudt.edu.cn (Y.L.); 3Graduate School of China Academy of Engineering Physics, Beijing 100193, China

**Keywords:** dense plasma, transient spatial localization, electron scattering cross sections, electron transport in gases, non-thermal plasma

## Abstract

Recent experiments have observed much higher electron–ion collisional ionization cross sections and rates in dense plasmas than predicted by the current standard atomic collision theory, including the plasma screening effect. We suggest that the use of (distorted) plane waves for incident and scattered electrons is not adequate to describe the dissipation that occurs during the ionization event. Random collisions with free electrons and ions in plasma cause electron matter waves to lose their phase, which results in the partial decoherence of incident and scattered electrons. Such a plasma-induced transient spatial localization of the continuum electron states significantly modifies the wave functions of continuum electrons, resulting in a strong enhancement of the electron–ion collisional ionization of ions in plasma compared to isolated ions. Here, we develop a theoretical formulation to calculate the differential and integral cross sections by incorporating the effects of plasma screening and transient spatial localization. The approach is then used to investigate the electron-impact ionization of ions in solid-density magnesium plasma, yielding results that are consistent with experiments. In dense plasma, the correlation of continuum electron energies is modified, and the integral cross sections and rates increase considerably. For the ionization of Mg${}^{9+}\;e+1s^{2}\;2s{\;}^{2\;}S\to 1s^{2}{\;}^{1\;}S+2e$, the ionization cross sections increase several-fold, and the rates increase by one order of magnitude. Our findings provide new insight into collisional ionization and three-body recombination and may aid investigations of the transport properties and nonequilibrium evolution of dense plasma.

## 1. Introduction

The electron-impact ionization processes of atoms and ions embedded in dense plasma are of interest both for fundamental science and for practical applications in the interiors of stars, including white dwarf and neutron stars [1,2], as well as for binary black holes, inertial confinement fusion [3,4], and high-energy-density plasmas [5,6]. They determine the charge-state distribution and power balance and can also be used to infer the electron temperature and electron density of plasma [7,8,9]. Electron scattering cross sections are required in order to model electron damage and to simulate single electron tracks into biological media [10,11]. To date, most atomic data on electron–ion collisional ionization have been obtained for isolated atoms or ions [7,12,13,14], where the coupling with the plasma environment was assumed to be trivial and was thus neglected. However, these data are only applicable to rarefied plasmas, for which there is, in general, a reasonable agreement between experimental and theoretical cross sections due to the adequate consideration of physical effects, such as by including enough excitation auto-ionization channels in the calculations [15,16]. If the plasma density is sufficiently high and the coupling cannot be neglected, plasma screening plays a non-negligible role and should be considered when determining the electronic structure. The effects of plasma screening on electron-impact ionization cross sections [17,18] have been extensively investigated using various analytical models [19,20,21,22].

Whether for isolated or screened ions, the present standard atomic collision theory describes incident and scattered electrons in electron–ion collisional ionization as distorted plane waves [23]. In such a treatment, the continuum electrons are completely coherent throughout the entire physical space during the process. In dense plasma, however, the continuum electrons collide randomly with the surrounding free electrons and ions, which results in a loss of phase of the electron matter wave and partial decoherence when the electron leaves the parent ion [24]. The partial decoherence of the continuum electrons is a natural many-body effect occurring in dense plasma and differs from plasma screening. The latter supplies an additional Hermitian potential to the physical system, whereas the former is not Hermitian and thus represents a type of dissipation mechanism in dense plasma. This work investigates how decoherence affects electron–ion collisional ionization processes.

Elusive signs may be found of anomalous behavior in continuum atomic processes in dense plasma [25,26,27,28,29,30]. In the solar interior, the internal structure predicted by the standard solar model [25] is inconsistent with helioseismic observations; this inconsistency could be resolved if the Rosseland mean opacity was increased by 10–20% [26,27]. Confined micro-explosion experiments using focused femtosecond laser pulses show that ion separation is enhanced in nonideal plasma [29]. Electron transport in high-plasma-density inductively coupled discharges displays a variety of nonequilibrium characteristics, and the skin depth is anomalous [30]. Only recently has it become feasible to experimentally investigate these processes [31,32]. Vinko et al. [31] studied the femtosecond collisional ionization rates in solid-density aluminum plasma created by the Linac Coherent Light Source [33], tuned to specific interaction pathways around the absorption edges of ionic charge states. Berg et al. [32] reported the first experimental measurements of ultrafast electron-impact ionization dynamics, for which they used resonant core-hole spectroscopy in a solid-density magnesium plasma. By resonance-pumping the 1s→2p transition in highly charged ions within an optically thin plasma, they measured how off-resonance charge states are populated via collisional processes on femtosecond time scales. These experimental results cannot be explained by the present standard atomic collision theory, even if the plasma screening effect is included.

The authors of [31,32] state that the collisional ionization and recombination cross sections are considerably larger than the values predicted by the present widely used models. The ionization cross sections and rates may be increased by including plasma screening, but they are not large enough to explain the discrepancies. These experiments provide new insight into ionization in dense plasma; nevertheless, understanding the physics and collisional mechanisms behind this enhancement remains a great challenge.

The goal of the present work is to understand the collisional ionization processes occurring in dense plasma and to explain the discrepancies between the present theory and experimental results. We point out that the transient spatial localization of continuum electrons (TSLCE) in collisional ionization enhances the cross sections and rates; thus, it is an important mechanism modifying electron-impact ionizations in dense plasma. Continuum one-electron-state localization is a natural many-body effect occurring in dense plasma and is different from plasma screening. We develop a theory to investigate how one-electron-state localization affects collisional cross sections and rates in both local thermodynamic equilibrium (LTE) plasmas and non-LTE plasmas. The theory is then applied to analyze the electron-impact ionization processes of ions embedded in solid-density magnesium plasmas.

The proposed theory predicts features in the compared energy differential cross sections that are not predicted by existing isolated- or screened-ion models, and these features significantly modify the collisional ionization dynamics. Thus, the theory considerably increases the integral cross sections and rates. This theory can be widely applied to studies of dense plasma, inertial confinement fusion, astrophysics, and high-energy-density physics. It also deepens our understanding of atomic properties within dense plasmas, which cannot be achieved with current theories. However, obtaining a unified, complete, and consistent treatment of the anomalous behavior of electron–ion collisional processes based on quantum many-body theory remains a challenge.

## 2. Results

### 2.1. New Features of Energy Differential Cross Sections

To begin our discussion, we first consider collisional processes in LTE plasmas. The electron-impact ionization in dense plasma exhibits features that are completely different from those of isolated ions. Figure 1 shows the energy differential cross sections for the ionization pathway $e+1s^{2}2s{\;}^{2\;}S\to 1s^{2}{\;}^{1\;}S+2e$ of Mg${}^{9+}$ in a solid-density magnesium plasma at an electron temperature of 150 eV with residual energies of 50, 200, 800, and 1400 eV, respectively. For a solid-density magnesium plasma at 150 eV, the free electron density is about $4.54\times 10^{23}$ cm${}^{-3}$, based on the ionization balance of the plasma. Three sets of results are calculated to compare the effects of plasma screening and of the TSLCE: results for an isolated ion, an ion in plasma with screening, and an ion in plasma with screening plus TSLCE. It can be seen that the inclusion of plasma screening increases the differential cross section by several-fold compared with that of the isolated ion. With the added consideration of TSLCE under the given plasma conditions, the differential cross section is further increased by about one order of magnitude with respect to that of an isolated ion.

Looking at the results in more detail, we find that, surprisingly, the differential cross section is modified by electron localization in a dense plasma; thus, it exhibits new features. The isolated-ion model predicts a “U-shaped” energy dependence for the energy differential cross section that is steeper at high residual electron energies [34,35,36,37,38,39]. The screened-ion model does not significantly modify this energy dependence, although the differential cross section exhibits some notable differences. If we include the effect of electron localization, however, the U-shaped energy dependence is only approximately maintained at low and high residual energies [see Figure 1a at 50 eV and Figure 1d at 1400 eV]. At an intermediate residual energy, the energy differential cross section differs completely from the isolated- and screened-ion models. At a residual energy of 200 eV [Figure 1b], the differential cross section features two minima, for a single ejected electron, at energies of ∼40 eV and ∼160 eV. With an equal sharing of energy for two continuum electrons (100 eV), the differential cross section features a local maximum, which should be a minimum in the isolated- and screened-ion models. At a residual energy of 800 eV [Figure 1c], we find a nearly equal differential cross section, except at the two ends, with one continuum electron capturing a small fraction of the residual energy and the other capturing the remaining residual energy.

This is a very surprising and counterintuitive result. These qualitative changes in the energy differential cross section are closely related to localization-induced changes in the continuum electron state, which evidently modifies how the one-electron ionization transition amplitude depends on the continuum electron energy. For an isolated ion, the one-electron ionization transition amplitude decreases monotonically from the ionization threshold to an energy level above that of the continuum electron, and this general behavior produces the U-shaped energy dependence of the energy differential cross section of the isolated ion. The localization of the continuum electron state and the broadening of the ionization threshold make the energy dependence of the one-electron ionization transition amplitude more complex: initially, it increases, and then it decreases at higher energies. Therefore, the product of the two-electron transition amplitude does not have a U-shaped energy dependence (when plotted as a function of the total energy of the two electrons) but rather a much more complex shape.

### 2.2. TSLCE-Induced Enhancement of Integral Cross Sections

Figure 2 shows the integral cross sections for the ionization channel of Mg${}^{9+}$$e+1s^{2}2s{\;}^{2\;}S\to 1s^{2}{\;}^{1\;}S+2e$ in a solid-density magnesium plasma at electron temperatures of 50, 100, 150, 200, and 250 eV. The most striking conclusion drawn from this figure is the significant increase in the integral cross section caused by the use of the TSLCE. In fact, the plasma screening enhances the collisional ionization processes, but the increase in the integral cross section due to TSLCE is much larger, in particular at a higher electron density. Here, we only compare the peak cross section to obtain a quantitative understanding. The peak cross section is calculated to be 0.106 Mb for an isolated ion, but this value is increased by 160% and 195% due to plasma screening at the given lowest (50 eV) and highest (250 eV) temperatures, respectively. With the TSLCE, the peak cross section is further increased by 4.90, 6.88, 9.80, 12.00, and 15.00 times at 50, 100, 150, 200, and 250 eV, respectively. The enhancement of the integral cross section caused by the plasma screening depends only weakly on plasma temperature. However, the TSLCE effect depends much more strongly on electron density and temperature. Moreover, the TSLCE has a more pronounced influence near, yet somewhat above, the ionization threshold, whereas the effect of plasma screening remains roughly similar over a much wider energy range of the incident electron.

The next striking conclusion is the evident ionization potential depression (IPD) caused by plasma screening. The ionization potentials of the 2s electron are predicted to be 198.28, 203.89, 206.99, 209.00, and 210.37 eV at plasma temperatures of 50, 100, 150, 200, and 250 eV, respectively, which is considerably less than our calculated value of 367.22 eV and the experimental value [40] of 367.49 eV for an isolated ion. Note that we consider energy broadening for the ionization threshold by including the TSLCE. The result is that the cross section near the ionization threshold behaves differently with and without the energy broadening. Finally, if the TSLCE effect is considered, the integral cross section grows more rapidly from the ionization threshold to a peak value and descends more rapidly. However, with the different treatments of the plasma effects, the predicted location of the peak energy for an isolated ion differs from that for an ion in plasma. When the TSLCE is included, the peak shifts to a lower incident electron energy.

The new features of the differential and integral cross sections of the ions in dense plasma are closely related to the TSLCE of electrons involved in electron-impact ionization. Momentum broadening is an important determinant of localization. Figure 3 shows the momentum broadening as a function of the energy of a continuum electron at plasma temperatures of 50, 100, 150, 200, and 250 eV. The momentum broadening increases rapidly from the ionization threshold up to a peak value, and then decreases much more rapidly with increasing continuum electron energy. This dependence of momentum broadening on energy explains the new features shown in Figure 1 and Figure 2. The energy differential cross section shows the largest modification compared with the isolated and screened ions at a residual energy of 200 eV [Figure 1b], which is a natural result of the TSLCE because the momentum broadening is large near this energy level. However, the TSLCE exerts much less effect at the residual energies of 50 eV [Figure 1a] and 1400 eV [Figure 1d] because of the smaller momentum broadening. Similarly, the pronounced enhancement of the integral cross section in a definite energy range near the threshold shown in Figure 2 also results from momentum broadening.

### 2.3. Greatly Increased Collisional Ionization Rates in Dense Plasmas

Figure 4 shows how the TSLCE affects the rates for the electron-impact ionization of $e+1s^{2}\;2s{\;}^{2\;}S\to 1s^{2}{\;}^{1\;}S+2e$ of an Mg${}^{9+}$ ion in solid-density magnesium plasma. The results indicate a significant enhancement of the electron–ion collisional ionization rates in dense plasma due to plasma screening and the TSLCE. At a lower plasma temperature, plasma screening plays a more important role than the TSLCE in this enhancement. The ratio of the rate obtained by considering plasma screening to that obtained by using the isolated-ion cross section is far greater than the ratio of the rate obtained by using the cross section of the screened ion without the TSLCE to that obtained with the TSLCE. At a higher plasma temperature, however, this ratio decreases, and the TSLCE plays an increasingly important role in enhancing the rates. Compared with the results obtained using the isolated-ion cross section, the rates increase by 64.0, 10.8, 5.5, 3.8, and 3.1 times due to plasma screening and by 177.6, 33.6, 22.1, 19.6, and 19.5 times due to both plasma screening and the TSLCE at plasma temperatures of 50, 100, 150, 200, and 250 eV, respectively.

The larger electron–ion collisional ionization rates in dense plasma are due to two factors: the increase in the integral cross section and the shift of the ionization threshold. At a lower temperature, the latter plays a more important role, so the IPD causes a greater enhancement of the electron–ion collisional ionization rates. The statistical weight from the energy distribution of the free electrons in the plasma is greater for the cross section with an IPD than without. As a result, the electron–ion collisional ionization rate increases more at lower temperatures, but the effect becomes much less profound with increasing temperatures. With increasing temperatures, the enhancement of the integral cross section caused by the TSLCE plays a more important role in increasing the rates.

### 2.4. Comparison with Experiments and Other Theoretical Results

We now test the proposed theory by comparing its results with recent experimental and theoretical results. The experiment of Berg et al. [32] used an X-ray free-electron laser to produce a solid-density magnesium plasma that strongly deviates from LTE and has a considerable density of core-hole states. Because the laser pulse is short (60 fs) and electron–ion collisional ionization rates are very fast, the free electrons in the plasma reach only an approximate equilibrium distribution, whereas the ions remain cold during the entire X-ray interaction.

These results refer to LTE plasma; here, we extend our approach to non-LTE plasma. To simplify the treatment, we assume a two-temperature plasma model with a definite electron density and temperature and a (different) definite ion temperature. The electron temperature and density are assumed to be 75 eV and $3.0\times 10^{23}$ cm${}^{-3}$, which are the same physical conditions considered by Berg et al. [32]. Figure 5 compares our calculated integral cross section with the results of experiments [32] and with other theoretical results for the collisional ionization $e+1s\;2s^{2}\;2p^{2}{\;}^{4\;}P\to 1s\;2s^{2}\;2p{\;}^{3\;}P+2e$ of Mg${}^{7+}$. These other theoretical results were obtained by Berg et al. [32] using different methods, including those of Lotz [41] and the revisions by Burgess and Chidichimo (BC) [42] and Dere [43]; the scaled hydrogenic approximation [44] and its analytic fitting formula by Clark, Abdallah, and Mann [45]; binary encounters [46,47,48,49]; and the distorted-wave approximation [50,51,52,53]. More recent updates regarding the electron-impact ionization processes of atoms and ions of H to Zn can be found in [54].

To explain their experimental results [32], Berg et al. proposed a parameterized expression for the collisional cross section (referred to by them as the “BCF with IPD” model) that aimed to describe the available experimental data in the low-density regime and provide a scaling method to capture the lowering continuum. Thus, the BCF model (with and without the IPD effect) obtained a consistent result (the same profile and trend) both in the low- and high-density regimes. Figure 5 shows that the predicted enhancement in the cross section obtained by including both screening and the TSLCE is consistent with the experimental result [32], although the very-near-threshold behavior differs. The results of the isolated-ion model under-estimate the cross sections by about one order of magnitude. Even considering plasma screening, the cross section values remain 2.0 to 5.0 times lower than the experimental data. The results in Figure 5 indicate that the TSLCE is at the origin of the enhanced collisional cross sections and rates observed experimentally [32].

## 3. Discussion

The TSLCE plays an important role in determining the differential and integral cross sections of electron-impact ionization. The localization that occurs in dense plasma differs from that of plasma screening. Although both TSLCE and plasma screening originate from many-body effects in plasma, their physical natures and origins differ. The former arises from the dephasing of the continuum electrons induced by multiple collisions with plasma particles, and the latter arises from Coulomb screening between particles and their environment. Thus, plasma screening affects both bound and continuum states, whereas TSLCE affects only continuum states. The new features in differential cross sections are attributed to electron localization (Figure 1) and indicate that the collisional dynamics and the electron correlation characteristics of electron-impact ionization differ completely from that of isolated ions. Figure 5 shows that TSLCE also modifies the energy dependence of the integral cross sections; in particular, at levels near but somewhat above the ionization threshold, where the collisional cross section increases less steeply than that of the BCF model. We conclude that TSLCE is a significant physical mechanism that enhances the cross sections and rates of ion ionization in dense plasma, as observed in experiments [31,32]. It is expected that the present theoretical formalism will find applications in the investigations of non-thermal plasmas in biological systems [55,56].

Two other types of localized electron states that occur in electron collisions with atoms or ions are similar to TSLCE: the shape resonance state and the Feshbach resonance state [57]. In a shape resonance state, the amplitude of the continuum electron wave function inside the centrifugal potential barrier is strongly enhanced. In a Feshbach resonance, the coupling between the open and closed channels induces wave-function mixing between the bound electrons and the continuum electrons of the open channel, resulting in a strongly enhanced amplitude of the continuum electron wave function inside the atomic or ionic sphere. Both the shape resonance state and the Feshbach-type resonance state increase the collisional cross section many-fold, or even by many orders of magnitude, over the nearby nonresonance states. However, the difference is that, for the shape resonance states and the Feshbach resonance states, the large increase in cross sections only occurs very near the resonance energy, whereas the TSLCE effect increases the cross section considerably over a much wider range of collision energies and increases the collisional ionization rate much more effectively.

From the comparison of the experimental and theoretical results of electron-impact ionization cross sections of Mg${}^{9+}$ embedded in a solid-density plasma, we find that the TSLCE can increase the cross section by 2.0 to 5.0 times compared with that obtained by including only the plasma screening effect. Practical calculations indicate that the TSLCE effect can be safely ignored below levels 0.001 times the solid density; however, with an increase in density, this effect becomes stronger and stronger up to a critical value of density. In the accretion disks of black holes [58], the TSLCE effect should be considered when obtaining atomic data, including electron-impact ionization cross sections and Auger widths [59].

## 4. Methods

### 4.1. Plasma Screening Potential

For an isolated ion of nuclear number *Z* with *N* electrons (N=Z refers to an atom), the atomic properties and wave functions are determined by solving the Dirac equation using a consistent field method [60,61]. In dense plasma, which is characterized by electron temperature *T* and density $n_{e}$, both the bound and continuum electrons of the ion feel a plasma screening potential $V_{scr}\left(r\right)$ that originates from the interaction with surrounding electrons and ions in the plasma [62,63,64,65,66]:(1)$$V_{scr}\left(r\right)=4\pi \left[\frac{1}{r}\int_{0}^{r}r_{1}+\int_{r}^{R_{0}}\right]r_{1}\rho \left(r_{1}\right)dr_{1}-\frac{3}{2}{\left[\frac{3}{\pi }\rho \left(r\right)\right]}^{1/3},$$
where $R_{0}={\left(\frac{3}{4\pi n_{i}}\right)}^{1/3}$ defines the radius of the ion sphere. The free-electron density distribution ρ(r) follows Fermi–Dirac statistics, which are obtained from the ionization equilibrium equation of the plasma. In the theoretical treatment, we calculate the Kohn–Sham potential by using the finite-temperature exchange correlation functionals of Dharma-Wardana and Taylor [67], although other choices for functionals are possible [68].

### 4.2. Localized Wave Functions of Continuum Electrons

The energy and momentum of the continuum electrons in dense plasma have a range of uncertainty because of the randomness of collisions and the resulting energy and momentum that is exchanged with other electrons and ions. As a result, the radial wave function of the continuum electrons (with central momentum $k_{0}$) is a superposition of the normalized wave function $P_{k\kappa }\left(r\right)$ associated with energy within the uncertainty [24,69]
(2)$$u_{k_{0}\kappa }\left(r\right)=\frac{1}{A}\int_{0}^{\infty }f\left(k,k_{0}\right)P_{k\kappa }\left(r\right)dk,$$
where κ is the relativistic angular quantum number, *A* is a renormalization constant, and $f(k,k_{0})$ is the expansion coefficient of $P_{k\kappa }\left(r\right)$ in momentum space. The normalized wave function $P_{k\kappa }\left(r\right)$ is determined by solving the Dirac equation with the plasma screening potential and has a definite energy and momentum *k*. In dense plasmas, the dominant momentum broadening has contributions from elastic and inelastic collisions with free electrons and ions in the plasma and, therefore, can be taken as a Lorentzian distribution $f\left(k,k_{0}\right)=\frac{\Delta k/\pi }{{\left(k-k_{0}\right)}^{2}+\Delta k^{2}}$ (with Δk being the half-width at the half-maximum of the momentum broadening). The localized wave function can be normalized to unity over the physical space domain [24].

### 4.3. Differential and Integral Cross Sections of Electron-Impact Ionization

The energy differential cross section for electron-impact ionization reads
(3)$$\begin{array}{cc} \frac{d\sigma_{if}\left(\epsilon_{0},\epsilon \right)}{d\epsilon }= & \;\rho \left(\epsilon_{0}\right)\rho \left(\epsilon \right)\rho \left(\epsilon_{0}-I-\epsilon \right)\frac{2\pi }{k_{i}^{2}g_{i}}\sum_{\kappa_{i}\kappa_{f}}\sum_{J_{T}}\left(2J_{T}+1\right)\\ \; & \times |\langle \psi_{i}\kappa_{i},J_{T}M_{T}|\sum_{p<q}\frac{1}{r_{pq}}|\psi_{f}\kappa_{1}\kappa_{2},J_{T}M_{T}{\rangle |}^{2}, \end{array}$$
where ρ(ϵ) is the density of states of the corresponding continuum electron [24], *I* is the ionization potential; $g_{i}$ is the statistical weight of the initial state; $k_{i}$ is the kinetic momentum of the incident electron; $J_{T}$ is the total angular momentum when the target state is coupled to the continuum orbital; $M_{T}$ is the projection of the total angular momentum; $\kappa_{i}$, $\kappa_{1}$, and $\kappa_{2}$ are the relativistic angular quantum numbers of the incident and scattered electrons; and $\psi_{i}$ and $\psi_{f}$ are the wave functions of the initial and final states, respectively. The density of states is included here to denote the re-normalization formalism of the continuum electron [24]. The integral ionization cross section is obtained by integrating the differential cross section over the energy ϵ of one scattered electron.

### 4.4. Collisional Ionization Rates

The rate of electron-impact ionization from level *i* to *j* may be expressed as [41]
(4)$$R_{ij}=n_{e}\int_{\sqrt{2I/m_{e}}}^{\infty }vf\left(v\right)\sigma_{if}dv,$$
where *I* is the ionization potential from *i* to *j*, and f(v) is the velocity distribution function of the free electrons in the plasma.

## 5. Conclusions

To summarize, electron–ion collisional ionization processes in dense plasma are strongly enhanced by TSLCE compared with those in rarified plasma. The dephasing and decoherence of the incident and scattered electrons significantly modify the wave functions of the continuum states and thus increase the electron–ion collisional ionization cross sections and rates. We develop a theory to account for the TSLCE effect and use it to investigate electron–ion collisional ionization processes in solid-density magnesium plasma.

The collision dynamics are completely modified at the residual energy level where the continuum electrons have a large momentum broadening and a two-peak structure appears in the energy differential cross section. The results of the proposed theory are consistent with the results of recent experiments. TSLCE influences not only the basic atomic processes but also the physical properties, including equation of state, opacity, electric conduction, and heat conduction. The results provide insight into the continuum atomic processes and their reverse processes (such as collisional ionization and three-body recombination, which are discussed in detail in this work) in the dense plasma regime. The proposed concept and theoretical formalism should find wide applications in a variety of disciplines, such as astrophysics, high-energy-density physics, inertial confinement fusion, and atomic and molecular physics. Further theoretical developments based on quantum many-body theory are required to obtain a unified and more natural and consistent description.

## Figures and Tables

**Figure 1 ijms-23-06033-f001:**
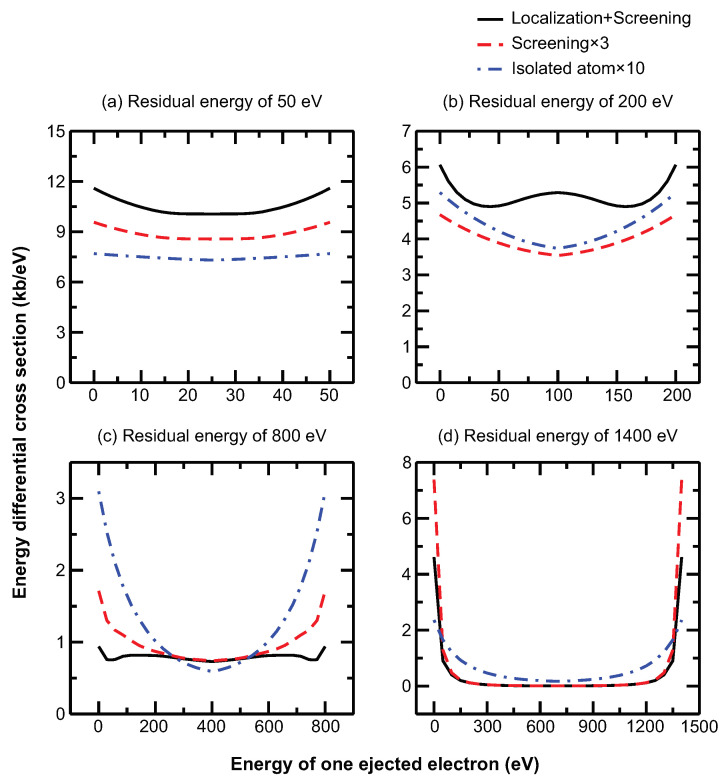
Energy differential cross sections of electron–ion collisional ionization. The ionization considered, *e* + $1s^{2}2s$${}^{2}S$→$1s^{2}$
${}^{1}S$ + 2*e* of Mg${}^{9+}$, occurs in a solid-density magnesium plasma at an electron temperature of 150 eV and a density of $4.54\times 10^{23}$ cm${}^{-3}$ with residual energies of (**a**) 50 eV, (**b**) 200 eV, (**c**) 800 eV, and (**d**) 1400 eV. For clarity, the results obtained with the isolated-ion and screened-ion models are scaled up tenfold and threefold, respectively.

**Figure 2 ijms-23-06033-f002:**
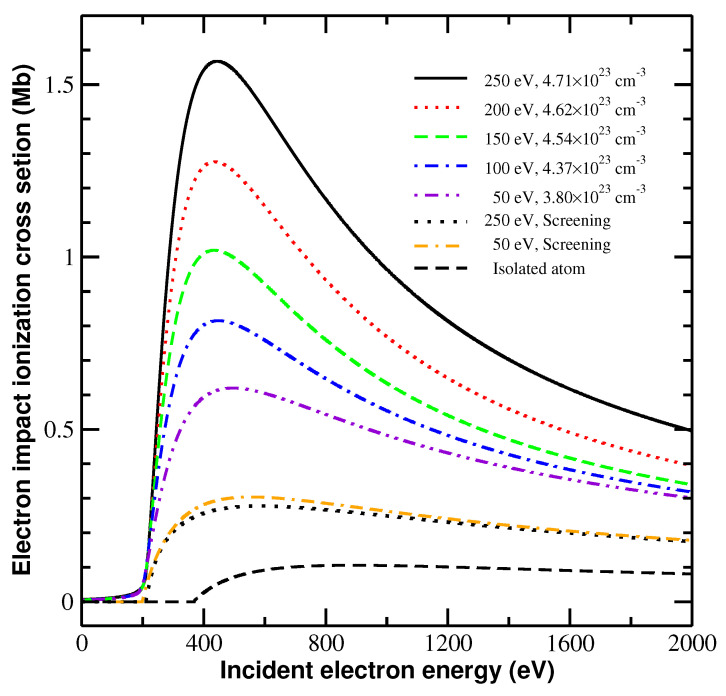
Enhanced collisional ionization integral cross section. The electron-impact ionization $e+1s^{2}\;2s{\;}^{2\;}S\to 1s^{2}{\;}^{1\;}S+2e$ of Mg${}^{9+}$ occurs in solid-density magnesium plasma at electron temperatures and densities of 250 eV and $4.71\times 10^{23}$ cm${}^{-3}$ (solid black line), 200 eV and $4.62\times 10^{23}$ cm${}^{-3}$ (dotted red line), 150 eV and $4.54\times 10^{23}$ cm${}^{-3}$ (dashed green line), 100 eV and $4.37\times 10^{23}$ cm${}^{-3}$ (dot-dashed blue line), and 50 eV and $3.80\times 10^{23}$ cm${}^{-3}$ (dot-dot-dashed violet line). The integral cross sections obtained by the screened-ion model are weakly temperature-dependent and, thus, for clarity, we only show the results at the highest (250 eV) and lowest (50 eV) temperatures.

**Figure 3 ijms-23-06033-f003:**
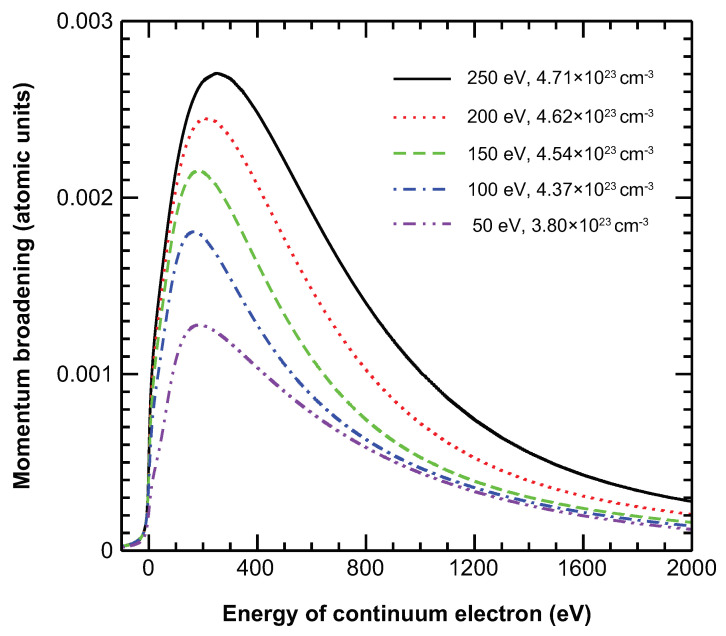
Momentum broadening of the continuum electrons. The continuum electrons are ejected from collisional ionization, $e+1s^{2}\;2s{\;}^{2\;}S\to 1s^{2}{\;}^{1\;}S+2e$ of Mg${}^{9+}$, occurring in a solid-density magnesium plasma at plasma temperatures of 250 eV (solid black line), 200 eV (dotted red line), 150 eV (dashed green line), 100 eV (dot-dashed blue line), and 50 eV (dot-dot-dashed violet line). Atomic units are used for the half-widths at half-maximum in the momentum distributions.

**Figure 4 ijms-23-06033-f004:**
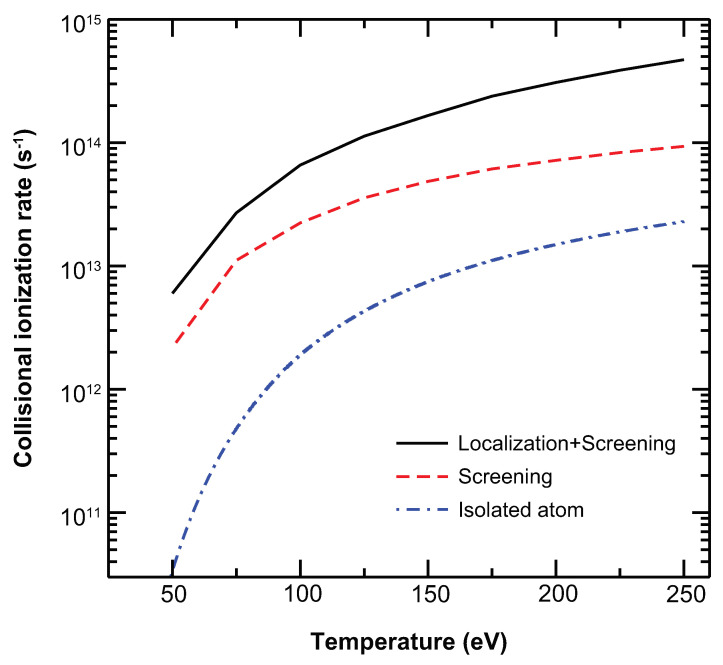
Electron–ion collisional ionization rates. The rates are given as a function of plasma temperature for solid-density magnesium plasma.

**Figure 5 ijms-23-06033-f005:**
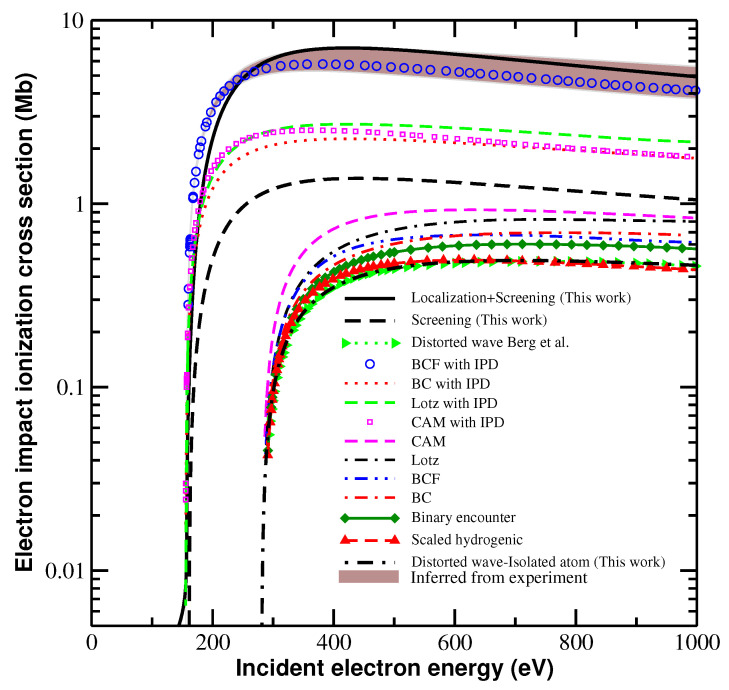
Collisional ionization integral cross section for core-hole state of Mg${}^{7+}$
$1s^{1}2s^{2}2p^{2}$. The electron-impact ionization considered, $e+1s\;2s^{2}\;2p^{2}{\;}^{2\;}P_{1/2}\to 1s\;2s^{2}\;2p+2e$ of Mg${}^{7+}$, occurs in solid-density magnesium plasma at an electron temperature of 75 eV and a density of $3.0\times 10^{23}$ cm${}^{-3}$. The cross sections we obtained are compared with the result inferred from the experiment by Berg et al. [32] and from theoretical calculations using Lotz [41] and the revisions by Burgess and Chidichimo (BC) [42]; the scaled hydrogenic approximation [44] and its analytic fitting formula by Clark, Abdallah, and Mann (CAM) [45]; binary encounters [46,47,48,49]; the distorted-wave approximation [50,51,52,53]; and the BCF model [32]. All data were taken from [32] except for the present work.

## Data Availability

The data that support the figures in this paper and other findings of our study are available upon request from the corresponding author.

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
