# Peer review of "The Strong Enhancement of Electron-Impact Ionization Processes in Dense Plasma by Transient Spatial Localization"

_ijms, 2022, doi:10.3390/ijms23116033_

Round 1

Reviewer 1 Report

  • A brief summary This work consists in including the phase loss of the electron matter wave and the partial decoherence of the continuum electrons, which are both effects that occur in dense plasma, into the calculations of the cross sections and rates of electron-impact-ionization processes. In particular, the case of Mg9+ + e was studied. The results seem to close the gap between measurements and previous calculations that are based on standard atomic collision theory.
  • General concept comments
    Article: The paper does not have major weaknesses. It fails to mention the density range within it the new calculations would apply and should be considered. The examples brought up were around 4E23/cm3, which is indeed a high-density plasma, but the authors can give at least a lower limit based on how well previous measurements agreed with the data from standard atomic collision theory. That will also improve the cited references. The possibility of applying the theory to biological media/system is mentioned at the beginning and toward the end, but it is difficult to perceive a high-density plasma (not the blood plasma) as an environment of molecules of biological interest, unless the definition of high-density plasma is given in values or numbers. It could be also appreciated if there is mention of the applications to black holes, including binary black holes.

Review: The manuscript is very interesting but could be improved. I would accept its publication after minor revision. The following are just suggestions as they are very minor and I would not require the authors to address those.

-        The cited references are too old for the data on electron-ion collisional ionization (1977, 1988, 2007, 2008). For instance, Urdampilleta et al. A&A 601, A85 (2017) could be added

-        Some papers tried to address the discrepancies between experimental and theories. It is worth to comment on those, at least in your introduction. See for example Kwon et al. APJ 784 (1), 13 published in 2014 and Kwon et al. PRA 86(2), 022701 published in 2012 or maybe others.

-        Lotz’s rate of electron-impact ionization was revisited by Dere A&A 466, 771-792 (2007).

-        I found the order of the sections very original and technical. However, putting section #4 before section #2 will allow the authors to refer the equations to their graphs.

  • Specific comments I would like the authors to address the followings and make change accordingly:

-        Line #23: Include the binary black holes.

-        Line #30-32: What is in number, the range of what you call high plasma density?

-        Line #71: I do not think it is appropriate to say that “it is a new physics”. It was just never been considered before. Please rephrase it.

-        Line #101: I do not understand the statement because I do not see through the graphs that inclusion of plasma screening CLEARLY lowers the IP. Or do you mean also that the consideration of TSLCE does lower the IP (by a lot)?

-        Line #137-141: There is no basis to compare high-density with low-density situation. I missed or maybe need more clarification.

-        Line #145: The “688%” suddenly break the number of significant digit rule.

-        Line #153-154: Where or how did you get the values of these IP? Also, the number of significant digits is again a problem (198.28 V vs 50 eV).

-        Line #163-164: In Fig. 2, the two dotted graphs (200 eV graph and 250 eV, Screening) look alike unless in color.

-        Line 201-204: Fig.4 shows rather increasing slopes only. It is just that the slope for the red one increases is the least—that slope does not show any decrease at all.

-        Line #210-211: In Fig. 5, including the references on each graph label will be helpful. For example, “BC [40]” etc. rather than just “using different methods [39-50]”

-        Line #252-258: This is rather part of your introduction. It does not discuss anything at all, or maybe reword this part to bring in the discussion.

-        Line # 281: Why did you choose this particular exchange-correlation functionals one versus the others? Would there be a big difference in the results?

Reviewer 2 Report

The manuscript entitled „Strong enhancement of electron-impact-ionization processes in dense plasma by transient spatial ionization” is motivated by the observation of much higher electron-ion collisional ionization cross sections and rates in dense plasmas than predicted by the most popular theories. To resolve that discrepancy authors developed a theoretical formulation to calculate the differential and integral cross sections by incorporating the effects of plasma screening and transient spatial localization. The manuscript is clearly written and in my opinion is suitable for publication in International Journal of Molecular Spectroscopy (MDPI). I have three specific comments and one very general:

Specific comments:

Figure 5: Ionization cross section values are plotted in logarithmic scale so the discrepancies between localization+screening and distorted wave-isolated atom is not visible.

Figure 5: the line style “dot dash dot” is used for four different theory levels – in color figure there is no problem but in the black and white version it is confusing

Figure 5 and also section 2.4 (Comparison with experiments and other theoretical results) – obtained cross sections are compared with the results from experiment [30] and from different theoretical calculations [39-50]. It should be clearly resolved which theoretical result is taken from which paper. Also used acronyms (like CAM, Lotz etc.) should be explained.

General comment:

The proposed mechanism is interesting and promising but In my opinion in really high density plasma, similarly to the case of charged particle interaction with atoms and molecules in the condensed phase) many multiple collisions can occur. Thus the description in terms of cross sections (which are always defined in single collision regime) can not be efficient and proper.
